# Predicting and improving diagnosis of tuberculosis outcomes in South Africa using machine learning techniques

Moses Asori[1,2], Desmond Mbe-Nyire Mpuure[3], Daniel Katey[4]*, Razak M. Gyasi[5,6,7]*

1 Department of Earth, Environmental and Geographical Sciences, University of North Carolina, Charlotte, United States of America, 2 Center for Applied Geographic Information Science, University of North Carolina, Charlotte, United States of America, 3 Department of Fundamentals of Economic Analysis, University of Alicante, Sant Vicent del Raspeig, Spain, 4 Trent Centre for Aging & Society, Trent University, 1600 West Bank Drive Peterborough, Ontario Canada, 5 Aging and Development Unit, African Population and Health Research Center, Nairobi, Kenya, 6 National Centre for Naturopathic Medicine, Faculty of Health, Southern Cross University, Lismore, New South Wales, Australia, 7 Department of Educational Leadership, Akenten Appiah-Menka University of Skills Training and Entrepreneurial Development, Kumasi, Ghana

* RGyasi.Research@gmail.com

## Abstract

The Ministry of Health and Social Welfare of South Africa has made significant efforts to combat tuberculosis (TB), guided by the National Strategic Plan for addressing HIV, STIs, and TB. However, progress in preventing and eradicating TB has been seriously hindered by reliance on ineffective diagnostic methods. This study aimed to predict and improve TB diagnosis in South Africa using machine learning techniques. Data from the National Income Dynamics Survey, conducted by the Southern African Labour and Development Research Units, were analyzed. The dataset underwent a 70:30 train-test split for Random Forest (RF), Decision Trees (DTs), Support Vector Machines (SVMs), Gradient Boosting Machines (GBMs), Artificial Neural Networks (ANNs), and Logistic Regression (LR). Hyperparameter tuning and impurity-based measures were employed to rank variable importance. RF achieved 87.50% sensitivity and an F1-score of 92.5%. DT achieved a sensitivity of 90.92% and an F1-score of 93.01%. ANN yielded 81.72% sensitivity and an F1-score of 87.53%. SGBMs showed 91.32% sensitivity and 94.55% F1-score. SVMs showed 90.03% sensitivity and 97.72% F1-score. LR achieved a sensitivity of 96.55% and an F1-score of 96.80%. Machine Learning (ML) techniques, with accuracy rates of more than 80% present a significant opportunity for enhancing TB prediction and diagnosis in South Africa. This predictive technique may be beneficial in resource-constrained settings, including those in sub-Saharan Africa.

## Introduction

Tuberculosis (TB) remains a significant global health challenge, with over 10 million new cases reported annually and approximately 1.7 million recorded deaths in 2022

**Data availability statement:** Data for the analysis is publicly and freely available via the following link: https://www.datafirst.uct.ac.za/dataportal/index.php/catalog/?page=1&sk=-NIDS&from=2008&to=2017&country%5B%5D=199&sort_by=title&sort_order=asc&ps=15.

**Funding:** The authors received no specific funding for this work.

**Competing interests:** The authors have declared that no competing interests exist.

[1]. Despite substantial progress in diagnosis and treatment, the complex dynamics of TB transmission and the variability in clinical manifestations pose significant obstacles to timely and accurate detection [2,3]. This challenge is particularly pronounced in the Southeast Asian and African regions, where factors such as co-infections, drug resistance, and social determinants of health compound the complexity [1], necessitating innovative approaches for effective disease control. In regions grappling with a high TB burden, including South Africa, the impact extends beyond individual health, permeating communities and straining healthcare systems [2–4]. Thus, achieving the global targets outlined in the *End TB Strategy* requires improved treatment and preventive measures, as well as a transformative shift in the approach to TB diagnosis [1].

Recent advancements in machine learning applications within the scope of TB diagnosis and treatment appear to offer a promising avenue for addressing TB prediction and diagnostic challenges in poor settings. Research has showcased the enhanced accuracy of deep learning in classifying X-ray images of potential TB patients, facilitating early detection and subtype differentiation [5]. Sharma et al. [6] explored TB drug resistance profiles using machine learning, guiding targeted treatment strategies. Acharya et al. [7] employed a progressive resizing approach to train models for automatic TB diagnosis, integrating fine-tuned Normalization-Free Networks (NFNets) and the Score-Cam algorithm to highlight specific regions in chest X-rays for detailed inference. This scholarly evidence highlights the practical application of machine learning and Artificial Intelligence (AI) in real-world clinical settings. As we navigate the evolving landscape of TB diagnosis, it becomes evident that machine learning stands at the forefront, offering innovative solutions and tangible advancements in our fight against this age-old global health threat. AI-based machine learning algorithms, such as naive Bayes, k-nearest neighbor, support vector machines, and artificial neural networks, have been used to predict how mycobacterium tuberculosis resistance in the genes rpoB, inhA, katG, pncA, gyrA, and gyrB for the drugs rifampicin, isoniazid, pyrazinamide, and fluoroquinolones, achieving an average accuracy of about 85% across all algorithms [8]. The performance of six AI-based machine learning models: decision tree (DT), artificial neural network (ANN), logistic regression (LR), radial basis function (RBF), Bayesian networks (BN), and support vector machine (SVM) have been developed, validated, and compared in their performance in predicting tuberculosis with DTs achieving the highest accuracy of 74.1% [9]. Similarly, RF, DT, naïve Bayes, and SVM have been used to predict pulmonary TB, with DT achieving the highest accuracy among all the models trained [10]. Beyond the clinical studies, Tang et al. [11] explored how meteorological and air quality factors influence TB incidence. They applied machine learning techniques, including BP (Backpropagation) neural networks, support vector regression, and random forest regression, to predict TB trends based on environmental factors like temperature, sunshine, and particulate matter. Similarly, Asad et al. [12] utilized machine learning to identify key factors contributing to TB treatment failure. By analyzing patient data from several high-burden countries, they employed feature selection techniques to identify the most significant demographic and medical factors influencing treatment outcomes.

South Africa remains a big name in contemporary TB-related reports and literature globally [2,4,13,14]. In 2020, the WHO updated the list of the top 30 TB-burdened countries globally, including South Africa [15]. Among these unique countries, it was discovered that South Africa had the third-highest absolute count of TB-reported cases, 172,200, and the fifth-highest prevalence of TB patients, 304,000 [15]. The country also assumed a leading role by having the highest number of TB cases concurrent with HIV, reaching 81,800, and appeared as the second-ranking country in the diagnosis of multidrug-resistant (MDR)-TB cases, with a significant count of 7,100 [16]. Although South Africa experienced some reductions in TB cases in 2023 [1], it appears that the country still bears a substantial share of the global TB burden, which necessitates innovative approaches to combat and potentially eradicate the disease effectively. Osman et al. [3] recently examined the systemic drivers of TB-related mortality in South Africa and offered valuable insights into the complex interplay between health system factors and their contribution to disease prevalence in the country. However, a critical gap in their analysis (probably due to their methodological approach) lies in exploring innovative technological interventions for improving TB diagnosis in the country. It is crucial to build on extant literature for several reasons: [1] to enhance the accuracy of detecting and identifying TB cases, [2] to ensure timely initiation of treatment for affected individuals, and [3] to advance research and development in the fight against TB.

Our study, therefore, aims to propose machine-learning methodologies for predicting and improving TB diagnostic outcomes in South Africa. Fundamentally, the study aims to bridge the gap between understanding the systemic drivers of TB mortality and implementing cutting-edge interventions that directly impact the diagnostic landscape, and to provide plausible recommendations to achieve global targets and milestones for reducing TB incidence and disease by 2030 [2].

## Materials and methods

### Data and sampling

We used data from the National Income Dynamics Survey (NIDS), a comprehensive survey conducted by the Southern African Labour and Development Research Units (SALDRU) in South Africa [17]. The survey employed a two-stage cluster sample design to sample households during the base wave. In the first stage, a sample of 400 primary sampling units (PSUs) was selected from the Master sample of 3000 PSUs used by Statistics South Africa for its Labour Force Survey and General Household Survey (LFSGHS) between 2004 and 2007. Notably, the NIDS PSUs constitute a subset of the Master sample. The 400 PSUs were randomly chosen from the strata of 52 district councils (DCs) [17]. NIDS, conducted every two years from 2008 to 2017, served as a nationwide panel survey that offered important insights into both individual and household characteristics. This longitudinal design enabled the tracking of household members, differentiating between continuing sample members (CSM) [Continuing Sample Member: All resident members of the original selected Wave 1 households or the Wave 5 Top-Up Sample (including children) and any children born to female CSMs in subsequent waves.] and temporary sample members (TSM) [Temporary Sample Member: A person who is not a CSM but is co-resident with a CSM at the time of the interview]. This rich dataset facilitated the exploration of dynamic household structures, changes in living conditions, and overall well-being.

Given the burden of diseases in South Africa, including sexually transmitted diseases, hypertension, tuberculosis, and stroke, among others, the NIDS is particularly valuable for understanding recent changes in the population's living conditions and their epidemiological profiles. We pool data from the last two waves of NIDS (2014/2015 and 2017) for analysis. This approach enabled us to provide policy suggestions based on recent trends. Our choice of surveys in the analysis aligned with our research objectives, regardless of whether the individual was a CSM or TSM. The attrition rate in the sample for the NIDS data was estimated to be approximately 18%, with high-income holders dropping out of the survey. The survey started its first round in 2008 with 28,000 residents of 7,303 households. This data had national coverage spanning all nine provinces and 52 districts of South Africa. Our study's final sample comprised the last two waves, representing 17,961 in 2014/2015 and 19,267 in 2017, resulting in a total sample of 37,228. This final sample was obtained after cleaning and considering the missing values of the relevant variables of interest.

## Outcome variable

Our outcome or target variable for training the ML models was TB. It was a binary categorical variable encoded as 0 or 1; 1 indicated those with TB, and 0 represented those without. Despite the significant strides in eliminating TB, it remains a significant epidemiologic burden to the country, partly due to a lack of precise diagnostic protocols. South Africa was ranked sixth globally for the high burden of TB (0.4-0.6 million) [18]. Some strategic targets set by the WHO in 2000 and 2015, such as halving the number of TB cases, have been met in South Africa [18]. However, the co-morbidity burden of TB, HIV/AIDS, and the increasing rate of non-communicable diseases continue to impede desired progress [19]. Even though the TB burden has attracted notable attention from the Ministry of Health and Social Welfare through the development of the National Strategic Plan (NSP) for tackling "HIV, STIs, and TB," precise predictive methods for its diagnosis have received limited research attention in the country and in the Sub-Saharan Africa (SSA) region in general. During the survey, respondents were asked whether they had been medically diagnosed with TB within the 30 days preceding the study. For example, respondents were asked, "We were previously told that you have Tuberculosis/ TB; is this correct?" "Have you ever been told by a doctor, nurse, or health care professional that you have Tuberculosis/ TB?" "Are you currently taking medication for Tuberculosis/ TB?" and "Do you still have Tuberculosis/ TB?"

## Predictors/features

After the multicollinearity test and feature selection process, sixteen predictors or features were used to train the ML models. They included blood cough, persistent cough, diabetes, diarrhea, high blood pressure, stroke, cancer, age, alcohol consumption, heart problems, fever, cigarette use, severe weight loss, self-rated health condition, and chest pain (See Table 1). All these variables were self-reported. As a country where there is a high confluence of infectious diseases, such as HIV/AIDS, TB, and many other chronic non-infectious diseases (CNDs), the health system is stressed, requiring innovative approaches to support resourceful decision-making.

TB risk is influenced by multiple risk factors, with notable ones being immunosuppressive diseases such as HIV/AIDS. For example, the risk of TB in South Africa has increased proportionally to the rate of increase in HIV/AIDS cases. Immune suppressive diseases can activate latent TB and catalyze its progression. Because HIV/AIDS status was not provided in the database, we used other proxies, including cancer, asthma, blood cough, and persistent cough, as suggestive elements for an immunocompromised individual or those with higher infectability tendencies. For example, one clinical study found that a 24-hour cough frequency was statistically significantly associated with TB-positive sputum smear status (p = 0.04) [20]. A recent review by Patterson and Wood [21] found that coughing alone is not necessarily a mechanical causal factor in transmission; it is linked to significant infectibility. Therefore, susceptible individuals, including family members, may easily contract TB through inhalation of airborne droplets [21]. Blood cough may also indicate advanced respiratory conditions such as pneumonia, which may suggest an immunocompromised situation [22]. These two factors were used as significant indicators for immunocompromised situations.

Also, severe weight loss, bloody sputum, persistent sweat/fever, and fatigue are substantial clinical symptoms of active TB [23,24]. Furthermore, an estimated 70% of people with diabetes now live in middle and low-income countries, with increasing rates in areas with high TB rates, such as countries in SSA [25]. Several reviews and primary studies have demonstrated an association between TB and diabetes. A meta-analysis of 13 studies found a 3-fold increase in TB among people with diabetes as compared to those without diabetes. Evidence suggests that having diabetes disrupts the innate and adaptive immune system, thereby affecting the body's ability to fight the TB bacterium effectively. Furthermore, several meta-analyses and systematic reviews have empirically demonstrated the relationships between TB and smoking. A systematic review by Bates and colleagues found a relative risk of 2.3–2.7 TB among smokers as compared to non-smokers [26]. A similar review found a 2-fold risk for TB among smokers as compared to non-smokers [25], suggesting that smoking remains a significant factor for TB development and transmission. Generally, a 2-fold increased risk of TB following alcohol consumption has also been found [27].

**Table 1. Predictor variables or features and descriptive statistics.**

| Variable | Observation | Proportion (%) |
|---|---|---|
| Fever | | |
| 0 | 29,492 | 79.1 |
| 1 | 7,736 | 20.9 |
| Tuberculosis | | |
| 0 | 35,847 | 96.3 |
| 1 | 1,381 | 3.7 |
| Persistent cough | | |
| 0 | 33,112 | 88.9 |
| 1 | 4,116 | 11.1 |
| Cough with Blood | | |
| 0 | 36,855 | 99.0 |
| 1 | 373 | 1.0 |
| Chest pain | | |
| 0 | 34,582 | 92.9 |
| 1 | 2,646 | 7.1 |
| Cancer | | |
| 0 | 36,767 | 99.1 |
| 1 | 323 | 0.9 |
| Hypertension | | |
| 0 | 32,760 | 88.0 |
| 1 | 4,468 | 12.0 |
| Asthma | | |
| 0 | 36,357 | 97.7 |
| 1 | 871 | 2.3 |
| Heart problem | | |
| 0 | 36,681 | 98.5 |
| 1 | 547 | 1.5 |
| Stroke | | |
| 0 | 36,977 | 99.3 |
| 1 | 251 | 0.7 |
| Diabetes | | |
| 0 | 35,860 | 97.2 |
| 1 | 1,052 | 2.9 |
| Diarrhea | | |
| 0 | 35,855 | 96.3 |
| 1 | 1,373 | 3.7 |
| Smoke | | |
| 0 | 30,471 | 81.9 |
| 1 | 6,757 | 18.1 |
| Severe weight loss | | |
| 0 | 36,122 | 97.0 |
| 1 | 1,106 | 3.0 |
| Alcohol consumption | | |
| Never | 11,680 | 31.37 |
| Rare | 8,649 | 23.23 |
| Drink | 16,899 | 45.39 |
| Self-rated health | | |
| Excellent | 12,059 | 32.39 |
| Very good | 11,763 | 31.60 |
| Good | 10,026 | 26.93 |
| Fair | 2,624 | 7.05 |
| Poor | 756 | 2.03 |
| Age | 37,228 | 35 |

Considering that a significant number of South Africans consume alcohol [28,29] and are smokers, they could be considered crucial to our TB prediction with ML. Age has also been found to be an important risk factor for TB, with increased probabilities in children and older adults. Age also moderates acquisition pathways. For example, one study found that children under two years acquired TB from home, whereas those over two years acquired it from sources within the community [30]. Thus, we included age as a feature in our prediction. Even though malnutrition has been shown as another risk factor for TB development, in our study, we used severe weight loss as a proxy for severe nutrition-associated wasting. We also considered high blood pressure based on systolic and diastolic blood pressures. High blood pressure was defined as a diastolic pressure of >90mm/hg and a systolic pressure of 140mm/hg. Although one systematic review found no significant link between TB and hypertension, it cautioned against interpreting the findings due to inherent limitations in their study design [31]. In contrast, one nationwide cohort study found a significant association between TB and hypertension [32]. Additionally, evidence exists for an association between diabetes and TB [33,34], which justifies its inclusion in training our models.

### Pre-processing

Pre-processing is crucial in ML, as insufficient data may generate bad results regardless of the robustness of the ML model. This study's pre-processing included handling missing data, correcting variable names, observing outliers, and transforming variables where necessary. Summary statistics and charts were used to explore the characteristics of our data and identify anomalies, including missing data, misspellings, and incorrect data formats. For this study, approximately 0.84% of the data points were missing. Assuming these data points were missing at random, we used multiple imputations using the chained equation (mice) technique in the R environment to impute the missing values based on the random forest technique. A bivariate regression analysis was conducted between TB and the predictors to avoid including unnecessary features for prediction. All predictors with a p-value >10% were excluded. The retained predictors listed above were subjected to the multi-collinearity test using the correlation matrix approach. None of the variables had a correlation coefficient >0.5 in the correlation matrix. Furthermore, these variables underwent a variance inflation factor (VIF) analysis using a binomial generalized linear model. None of them had a VIF > 5. Therefore, it was assumed that no multicollinearity existed that could threaten our model's adequacy. All our variables except age were categorical.

### Statistical analysis

ML is a non-parametric yet robust model for handling complex and non-linear patterns within data, making it more suitable for current data science tasks. It is a cyclical process that may evolve from data acquisition, preprocessing/data engineering/feature engineering, model calibration, and validation

A simplified overview is given in [Fig 1].

We used six ML techniques for our classification task. There are random forests (RF), decision trees (DT), Logistic Regression (LR), artificial neural networks (ANN), stochastic gradient boosting machines (SGBM), and support vector machines (SVM). We used a grid search approach to tune the hyperparameters (See S1 Text). Using the Friedman rank sum test, we compared the accuracies and kappa values of the models. There was a statistically significant difference when $p < 0.05$. For a detailed description of the technical aspects of the six models trained, including how they were set up, trained, and evaluated, (see S1 Text).

### Model performance assessment

Model validation, including its generalizability, is crucial in determining whether the model can be relied upon. This involves ensuring that the model can be applied to a broader population, like other African countries with similar sociodemographics. We used a confusion matrix to derive several performance metrics. These metrics assessed model adequacy and reliability, which is how well the model fits the data and generalizes to unseen data. Metrics included accuracy,

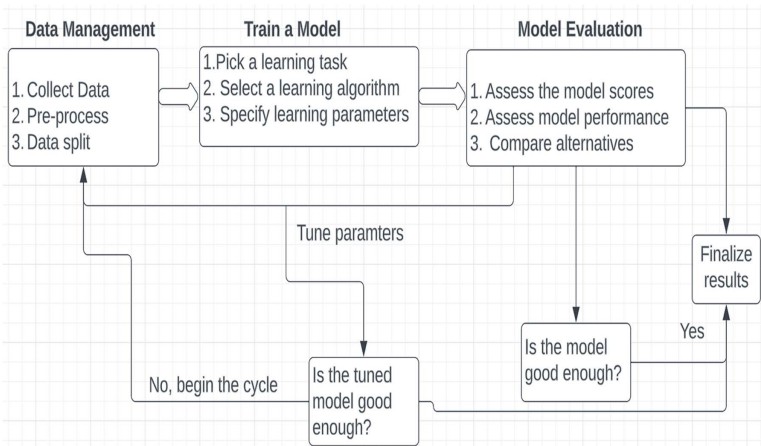

**Fig 1. An ML schema.**

sensitivity, specificity, F1-score, negative predictive value (NPV), and positive predictive value (PPV). These metrics are calculated as follows.

$$Accuracy = \frac{TP + TN}{TP + TN + FP + FN} \tag{1}$$

$$PPV = \frac{TP}{TP + TN} \tag{2}$$

$$Sensitivity = \frac{TP}{TP + FP} \tag{3}$$

$$Specificity = \frac{TN}{TN + FP} \tag{4}$$

$$NPV = \frac{TN}{TN + FN} \tag{5}$$

$$F1 = 2 * \frac{Precision * Recall}{Precision + Recall} \tag{6}$$

Where TP is the true positive, TN is the true negative, FP is the false positive, FN is the false negative case, and NPV is the Negative Predictive Value. And F1 is the F1 score. The area under the curve or receiver operating characteristic (ROC) was used to evaluate the predictive accuracy of the algorithm. It plots iterative TP versus FP values based on various cutoffs. The area under the curve (AUC) values suggests the ability of the algorithm to discriminate between TB-positive and TB-negative respondents and represents the area under the ROC. The F1-score was assessed in

conjunction with sensitivity and accuracy to determine the model's performance, as accuracy alone could provide misleading results due to the severe class imbalance (1:26).

## Results

### Summary characteristic

The summary characteristics of our predictor variables are shown in Table 1. A total of 37228 respondents were used to train our model. About 3.7% reported having been diagnosed with TB, with an average age of 35.09 years (95% CI: 34.92-35.26 years). Approximately 21% had a fever, and those with a persistent cough accounted for 11.1%, while those with a bloody cough made up 1.0%. Furthermore, those who reported chest pain were 7.1%, cancer was 0.9%, and hypertension was 12.0%. Asthma was reported by 2.3%, while 1.5% reported having heart problems. Those who reported having a stroke were 0.7%, whereas diabetic respondents were 2.9%. Those ever diagnosed with diarrhea were 3.7%, and smoking was 18.1%. With alcoholic consumption, those who have never taken alcohol were 31.37%, those who rarely drink were 23.23%, whereas those who are regular drinkers were 45.39%. Regarding self-rated health conditions, 32.39% rated their health as excellent, 31.60% rated it as very good, 26.93% rated it as good, 7.05% rated it as fair, and 2.03% rated it as poor.

### ML prediction results

Based on the confusion matrix, performance characteristics for RF were: accuracy was 89.08% (95% CI: 88.5, 89.66%), sensitivity was 91.50%, specificity was 32.45%, PPV was 96.99%, whereas NPV was 11%. We also found an F1-score of 92.50%. Concerning DT, accuracy was 86.87% (95% CI: 86.23, 87.49%), sensitivity was 88.95%, specificity was 33%, PPV was 97.17%, whereas NPV was 11%. Also, the F1-score was 93.01%. Concerning ANN, the accuracy was 78.20% (95% CI: 77.46, 79.00%), sensitivity was 79.41%, specificity was 47.83%, PPV was 97.51%, and NPV was 8.28%. The F1-score was 87.53%. Also, according to SGBM, the accuracy was 89.8% (95% CI: 89.22, 90.36%), sensitivity was 92.11%, specificity was 30.53%, PPV was 97.15%, whereas NPV was 13.07%. The results of the LR are as follows: accuracy was 93.84% (95% CI: 93.38, 94.28), a sensitivity of 96.55%, a specificity of 18.35%, a PPV of 97.05%, and an NPV of 16.03%. Lastly, the results of the SVM are as follows: accuracy was 95.5% (95% CI: 95.11, 95.88%), sensitivity was 99.01%, specificity was 5.05%, PPV was 96.4%, whereas NPV was 16.7% (See Table 2,3 and Fig 2). Based on the Friedman rank sum test, there were statistically significant differences in accuracy and Kappa (Friedman $\chi^2 = 20.88$, $p$-value = 0.0004994).

Blood cough consistently came top in all the ensemble models for variable importance. For example, in the RF model, blood cough was ranked first, followed by stroke, severe weight loss, chest pain, and persistent cough (See Fig 3). The last three variables that contributed least to the RF were hypertension, age, and cancer.

In DT, the top five most essential predictors in decreasing order of significance were heart problems, asthma, severe weight loss, stroke, and blood cough. However, the last three variables contributing the least were cancer, age, and alcohol. In an LR model, the top three predictors, in order of importance, were stroke, blood cough, and heart problem, while the last three were age, alcohol consumption, and diarrhea. In an ANN model, the top five most essential variables in decreasing order of importance were blood cough, stroke, asthma, severe weight loss, and hypertension. On the contrary, the three least critical variables were cancer, age, and self-rated health status. For SGBM, the top five most essential predictors in decreasing order of importance were blood cough, stroke, severe weight loss, asthma, and chest pain. The last three least important ones were hypertension, cancer, and alcohol (See Fig 3).

## Discussion

Tuberculosis remains a critical health challenge in developing countries, alongside HIV/AIDS and other non-communicable diseases (NCDs). In South Africa, the burden of TB on healthcare systems is substantial [19]. Even though

**Table 2. The confusion Matrix for various ML models.**

**Random Forest**

| | | Reference | |
|---|---|---|---|
| **Predictions** | | **Non-TB** | **TB** |
| | Non-TB | 9794 | 304 |
| | TB | 910 | 112 |

**Decision Trees**

| | | | |
|---|---|---|---|
| | Non-TB | 9521 | 277 |
| | TB | 1183 | 139 |

**Artificial Neural Network**

| | | | |
|---|---|---|---|
| | Non-TB | 8501 | 217 |
| | TB | 2203 | 199 |

**Stochastic Gradient Boosting Machines**

| | | | |
|---|---|---|---|
| | Non-TB | 9859 | 289 |
| | TB | 845 | 127 |

**Support Vector Machine**

| | | | |
|---|---|---|---|
| | Non-TBTB | 10599105 | 39521 |

| Ridge Logistic Regression | | | |
|---|---|---|---|
| | Non-TB | 10409 | 316 |
| | TB | 372 | 71 |

**Table 3. The performance metrics for various ML models.**

| | Random Forest | Decision Trees | Artificial Neural Network | Stochastic Gradient Boosting Machines | Support Vector Machine | Ridge Logistic Regression |
|---|---|---|---|---|---|---|
| **Metric Name** | | | | | | |
| **Accuracy** | 0.8908 [95% CI: 0.8849, 0.8966] | 0.8687[95% CI: 0.8623, 0.8749] | 0.782[95% CI: 0.7746, 0.79] | 0.898[95% CI: 0.8922, 0.9036] | 0.955[95% CI: 0.951, 0.958] | 0.938 (95% CI: (0.938, 0.942) |
| **Kappa** | 0.1184 | 0.1093 | 0.0827 | 0.1378 | 0.0612 | 0.1392 |
| **Sensitivity** | 0.9150 | 0.8895 | 0.7941 | 0.9211 | 0.9901 | 0.9655 |
| **Specificity** | 0.3245 | 0.3341 | 0.4783 | 0.3053 | 0.0504 | 0.1835 |
| **Pos Pred Value** | 0.9699 | 0.9717 | 0.9751 | 0.9715 | 0.9640 | 0.9705 |
| **Neg Pred Value** | 0.1096 | 0.1051 | 0.0828 | 0.1307 | 0.1666 | 0.1603 |
| **Prevalence** | 0.9626 | 0.9626 | 0.9625 | 0.9626 | 0.9625 | 0.9653 |
| **Detection Rate** | 0.8808 | 0.8811 | 0.7644 | 0.8866 | 0.9531 | 0.9320 |
| **Detection Prevalence** | 0.9081 | 0.8921 | 0.7839 | 0.9126 | 0.9886 | 0.9603 |
| **Balanced Accuracy** | 0.6035 | 0.6118 | 0.6362 | 0.6132 | 0.5203 | 0.5745 |
| **F1-Score** | 0.9250 | 0.9301 | 0.8753 | 0.9455 | 0.9772 | 0.9680 |
| **AUC** | 0.7100 | 0.6921 | 0.7056 | 0.7250 | 0.5988 | 0.7219 |
| **No Information Rate** | 0.9626 | 0.9626 | 0.9626 | 0.9626 | 0.9626 | 0.9653 |

Also, AUC for RF was 71.00%, DT was 70.00%, ANN was 71.00%, SGBM was 73.00, and SVM was found to be 60.00%. See below.

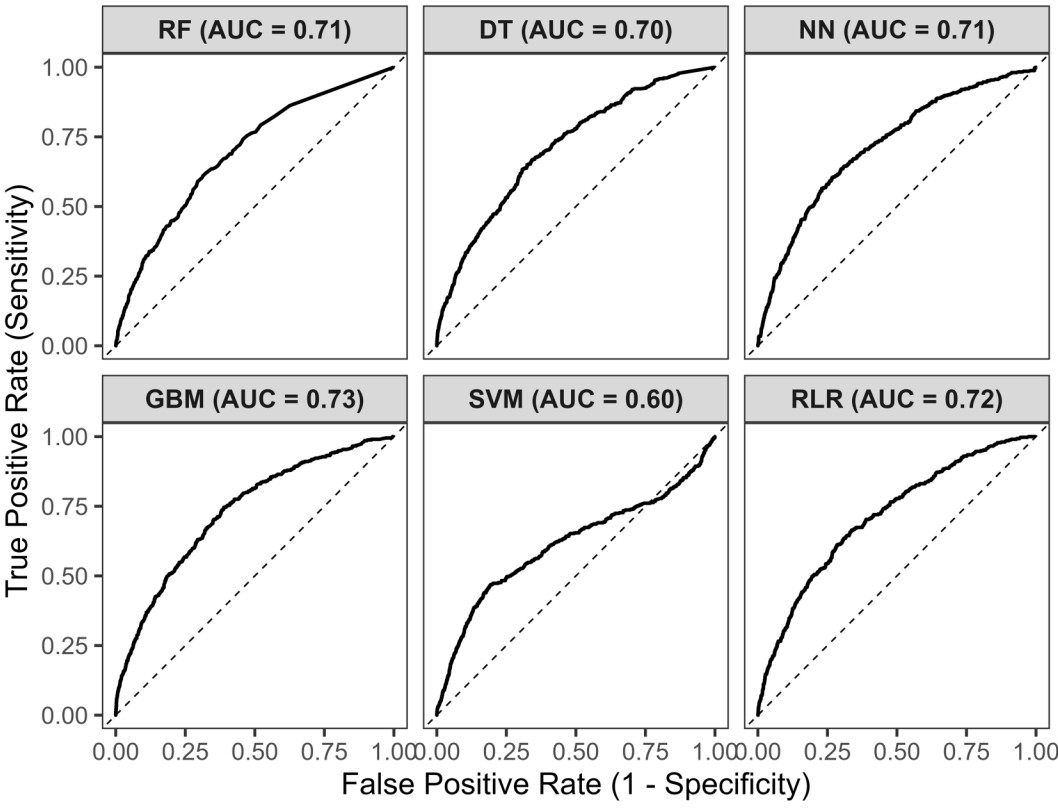

**Fig 2. Models performance metrics.**

the TB burden has attracted notable attention from the South African Ministry of Health and Social Welfare through the development of the National Strategic Plan (NSP) for tackling "HIV, STIs, and TB [35], preventive and eradication efforts have been stalled, partly due to outdated and inefficient diagnostic and predictive methods. Therefore, the importance of our current study cannot be overstated, especially in resource-constrained settings where the clinical diagnosis of TB and identification of underlying risk factors remain primarily limited. Using nationally representative data, five established machine learning models were employed to identify the most accurate model for enhancing TB diagnosis and prediction in clinical practice. Besides the merit of individual-level intervention, our study can shape system-based thinking regarding program designs and resource allocation [36]. Also, the process of facilitating corrective actions through reforming existing patient-oriented decisional frameworks may be enhanced. In this study, we have utilized sixteen non-invasive personal information items, including self-reported blood cough, persistent cough, diabetes, diarrhea, high blood pressure, stroke, cancer, age, alcohol consumption, heart problems, fever, cigarette use, severe weight loss, self-rated health condition, and chest pain, to discriminate TB status precisely among participants.

RF, DTs, SGBMs, ANN, and SVMs were used for the predictive exercise. Accounting for class imbalance through F1-score estimation, the models performed relatively well despite the extreme imbalance (1:26). We also found an AUC above 60% across all models (See Fig 4). However, RF and SGBMs, which comprised ensemble learners, as well as the Ridge LR model, outperformed neural networks with an F1-score above >92% and sensitivity above 91%. They were, on average, 13% higher than ANNs on the sensitivity metric and about 6% for the F1-score. Using an algorithmic process to discriminate clinical outcomes has several advantages over traditional statistics, such as [1] reducing the risk of functional misspecifications, [2] enhancing the handling of nonlinear structure within data, and [3] improving the

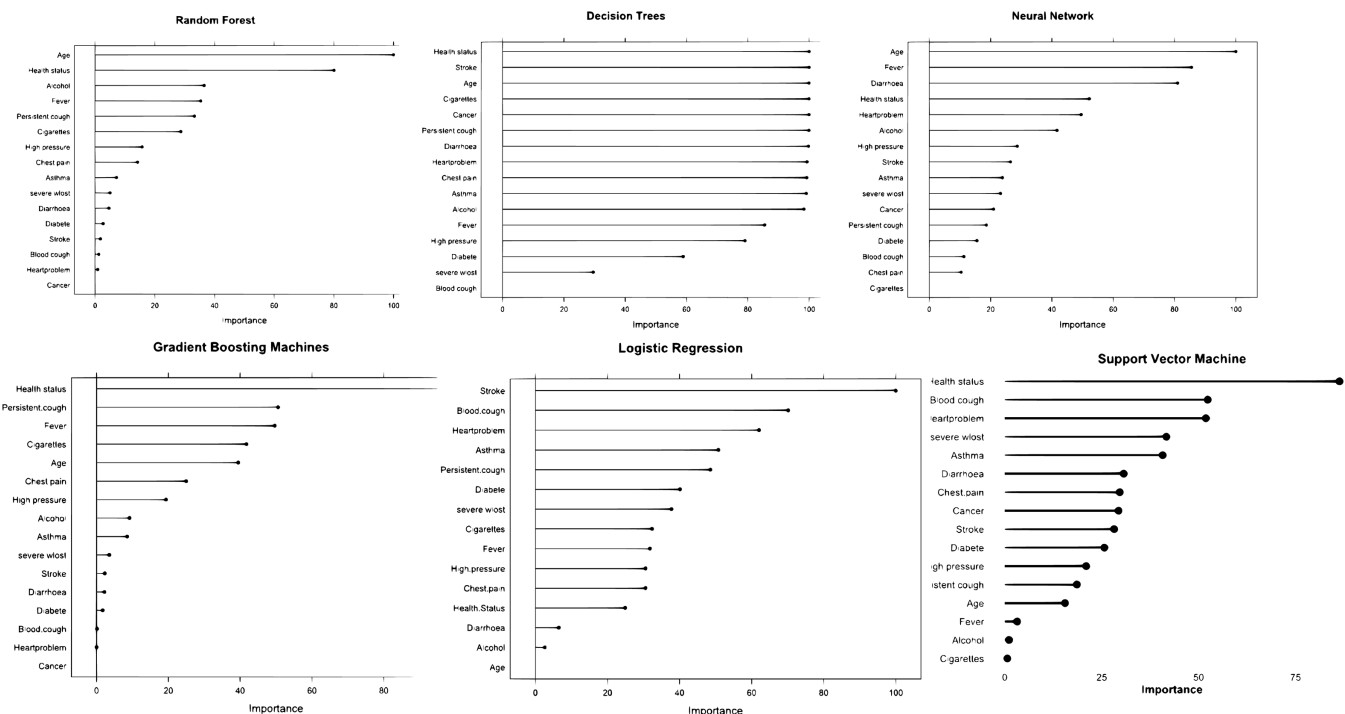

**Fig 3. Variable importance plot for the predictors/features.**

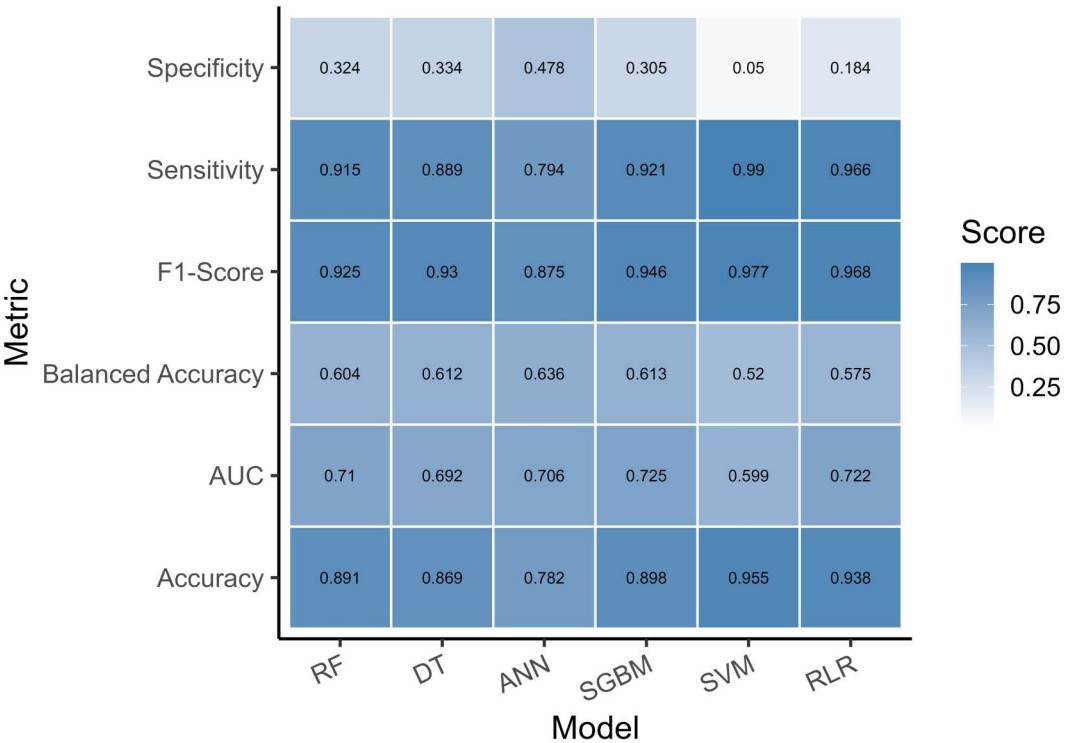

**Fig 4. Receiver Operating Characteristics for various ML models.**

ability to test for model generalization through model validation, which is challenging to achieve adequately in traditional statistical approaches. Unsurprisingly, AI and ML approaches are increasingly used to inform clinical and epidemiological practice. For example, studies, including the work of Yadav et al. [5], showcase the enhanced accuracy of deep learning in classifying X-ray images of potential TB patients, facilitating early detection and subtype differentiation. Furthermore, Sharma et al. [6] explored TB drug resistance profiles using machine learning, which guides targeted treatment strategies. Acharya et al. [7] employed a progressive resizing approach to train models for automatic TB diagnosis, integrating fine-tuned Normalization-Free Networks (NFNets) and the Score-Cam algorithm to highlight specific regions in chest X-rays for detailed inference.

Although there were high F1-scores and sensitivities, the specificities were relatively low, typical for disease cases with extreme skewness. Despite DTs exhibiting comparative accuracy as SGBMs, the latter demonstrated superior sensitivity and F1-score results, indicating the robustness of SGBMs. Of course, SGBMs are one of the most robust models presently, sometimes outperforming deep learning models; it is, therefore, not surprising that they will have such superiority over the single-tree decision model, which is prone to overfitting. Besides predictive outcomes, understanding the contribution of variables to model performance provides crucial insights for developing preventive strategies. For instance, identifying the most influential predictors for TB enables clinicians to target the most vulnerable epidemiological and demographic groups, facilitating more effective resource allocation. Across all the ensemble models (RF & SGBMs) and ANNs, persistent cough, health status, and age appeared influential in the differentiation of TB status among our participants. This finding corroborates other studies. For example, one clinical study found that a 24-hour cough frequency was statistically significantly associated with TB-positive sputum smear status (p = 0.04) [20]. A recent review by Patterson and Wood [21] found that coughing alone is not necessarily a mechanical causal factor in transmission; it is linked to significant infectibility. A bloody cough may also indicate an advanced level of pathogenic invasion and immunosuppression. Additionally, persistent sweating, fever, and fatigue are significant clinical symptoms of active TB [23,24]. Therefore, it is unsurprising that the combined outcome of these, which may define people's perceived health status, was also a significant feature in our predictive models. Additionally, all models found age to be an influential factor.

Even though differences in algorithmic pattern recognition and model complexity may lead to different variable importance rankings, the consistency of people's health status as the top three most influential predictors in most of our models suggests its criticality in the clinical identification of TB. Of course, this is expected as people facing multiple health complications, including communicable and non-communicable ones, will generally perceive their health status as poor relative to those who are free from these ailments. Also, stroke was the top predictor in the DT model. A complex-mediated immunosuppression following stroke may increase the risk of dysphagia and aspiration, which may elevate the risk of respiratory diseases such as TB. However, we need to emphasize that the linkage between TB and stroke is not straightforward: it is multifaceted and highly mediated by many other underlying health conditions, such as diabetes and hypertension. Therefore, causal interpretation in our study should be made with caution. We also found that chest pain and fever contributed moderately to our prediction across almost all models. It is, therefore, unstartling that the presence and severity of fever have been used to evaluate anti-TB drug effectiveness in some cases. One crucial finding was that, despite the use of the grid-search optimization technique, all the ensemble models outperformed both ANNs and SVMs. This may be explained by the presence of non-linearity in our data structure. For example, SVMs and ANN models may be less efficient in mathematically mapping predictors to the response variable(s) in non-linear patterns. Additionally, ensemble models like RFs and SGBMs are robust to noisy data, outliers, and small sample effects. Generally, our study demonstrates the potential of MLs in enhancing clinical assessment, identification, and prediction of TB in SSA, where its prevalence is still relatively high. In particular, several operationalization options available from the findings of this study include the implementation of digitization fusion for expedited screening, the establishment of a clinical decision support system, and the engagement of stakeholders through pilot testing. For example, given the lower computational burden and enhanced interpretability of robust models such as RF and Logistic Regression (LR), it may be beneficial to incorporate

these models within a mobile or tablet-based application system designed to aid community health workers in tuberculosis (TB) screening. Moreover, integrating sophisticated models, such as Gradient Boosting Machines (GBMs) and RFs, which are known for their high accuracy and sensitivity, into facility-level electronic health records may prove advantageous for clinical practitioners to identify high-risk TB patients, thereby enabling proactive response measures. To guarantee that these two application pathways are effectively implemented in demographically pertinent settings, further investigations dedicated to the development and piloting of these digital application platforms, facilitated by community and facility-based stakeholder engagement, may represent a promising approach, particularly in resource-constrained environments. These models are applicable at the screening stage, where [1] individuals can be identified for confirmatory testing, and [2] facility or geographic-level hotspots can be identified to flag active cases. Also, it makes it possible to conduct contact tracing. In low-income settings with adequate screening tools, case identification at the initial stage is based on symptom-based screening, which can be less accurate. ML models that integrate both demographic and clinical data have the potential to increase sensitivity and reduce the rate of missed cases. Also, with quantitative metrics including sensitivity, specificity, f1-score, and balanced accuracy, ML models provide you with objective performance measures to better identify which models provide the most optimal results. With further validation and addressing relevant limitations within a clinically pertinent setting, these models could be expanded and utilized to inform tuberculosis diagnosis across numerous countries in Africa experiencing similar challenges. To ensure that this decision support system more accurately reflects variations in epidemiological profiles across different countries and time periods, this study may serve as a benchmark for scalability through the incorporation of multi-center data and, potentially, the adjustment for geographic confounding factors that may restrict spatial generalizability.

## Limitations

The study relied on cross-sectional and self-reported data, which may have introduced recall biases and limitations in capturing the temporal aspects of disease conditions. However, by merging data from a two-period survey, we aimed to enhance the predictive power of our models. It is essential to acknowledge that machine learning models can perpetuate demographic biases in their predictions and algorithmic decisions, which may have significant implications for patient care and health decision-making processes. Due to data constraints, we were unable to train and validate our models across racial groups, particularly for underrepresented groups such as Whites and Asians, comprising less than 20% of South Africa's racial composition. Overall, the dataset exhibited a significant class imbalance. Conducting further stratified analysis based on race under such imbalance conditions would compromise the statistical reliability and robustness of the results, particularly in terms of accuracy, sensitivity, and generalizability. Therefore, we cannot be conclusive about the predictive nature of our models for these underrepresented groups. Additionally, gender-specific training and testing were not conducted for the reasons highlighted above. However, given the roughly equal gender distribution (a 45:55 ratio) in our dataset, we anticipate that the predictive accuracies and diagnostic guidance derived from our study can be reasonably applied to both males and females with minimal demographic bias. Future studies that collect and report more demographically balanced data, enabling robust subgroup analyses, will be crucial for the equitable development and deployment of algorithms in public health. Another important limitation of our study is the omission of HIV/AIDS status as a predictive variable. We lacked this information in our datasets, making it difficult to include. Since TB/HIV co-morbidity is a well-known challenge, especially in South Africa [37–39], excluding this co-infection factor may limit the generalizability of our models, especially in populations with varying HIV/AIDS prevalence. We recommend that future research include HIV/TB co-infection to improve accuracy and contextual relevance.

Again, we focused on the general TB status due to data constraints. A subgroup assessment on extra-pulmonary TB, which comprises 15% of TB cases in South Africa, was not conducted. Thus, extrapolation to specific types of TB should be done with caution. Lastly, due to the extreme class imbalance in the test dataset, which is typical for highly skewed data, the reported accuracy may be misleading, as it is primarily determined by the majority class (those without TB).

Readers, therefore, must pay more attention to multiple metrics, including sensitivity, specificity, F1-score, AUC, and balanced accuracy. While the models demonstrated comparatively higher sensitivity and F1-scores, their specificity was lower, which is a well-recognized challenge in the context of imbalanced clinical data. This limitation underscores the need to position such machine learning approaches as adjunctive decision-support tools rather than as replacements for clinical judgment. To minimize the potential consequences of false-positive classifications, model outputs should be integrated with established confirmatory diagnostics, including GeneXpert testing and chest radiography. Moreover, to ensure sustained reliability and adaptability in diverse healthcare settings, periodic recalibration and external validation of these models will be essential.

## Conclusions

TB continues to present ravaging effects on developing countries, and South Africa is no exception. To contribute to enhancing clinical practice through the prediction and diagnosis of TB, we developed five known ML models: RF, SVM, SGBMs, DTs, and ANN, and compared their efficiencies. We used sixteen non-intrusive personal attributes for the prediction. We found that tree-based models (SGBMs, DTs, and RFs) and L2 regularized LR outperformed ANN and SVMs. However, based on accuracy and sensitivity, all models achieved an accuracy and sensitivity of >70%, with SGBMs, LR, and RFs achieving satisfactory sensitivity, specificity, AUCs, and accuracy measures. Our study may be crucial to TB diagnostics, prevention, and management in the immediate and long term.

## Supporting information

**S1 Text. Supporting Information.**
(DOCX)

## Author contributions

**Conceptualization:** Moses Asori, Desmond Mbe-Nyire Mpuure.

**Data curation:** Moses Asori, Desmond Mbe-Nyire Mpuure, Daniel Katey.

**Formal analysis:** Moses Asori.

**Methodology:** Moses Asori.

**Software:** Moses Asori.

**Supervision:** Desmond Mbe-Nyire Mpuure, Daniel Katey, Razak M Gyasi.

**Writing – original draft:** Moses Asori, Desmond Mbe-Nyire Mpuure, Daniel Katey.

**Writing – review & editing:** Desmond Mbe-Nyire Mpuure, Daniel Katey, Razak M Gyasi.

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
