## [Decision Letter · Decision Letter 0]

14 Jan 2025

PGPH-D-24-02823

Predicting Tuberculosis (TB) with Machine Learning Approaches: An Effort to Improve Tuberculosis Diagnosis in South Africa

Dear Dr. Gyasi,

Thank you for submitting your manuscript to PLOS Global Public Health. After careful consideration, we feel that it has merit but does not fully meet PLOS Global Public Health’s publication criteria as it currently stands. Therefore, we invite you to submit a revised version of the manuscript that addresses the points raised during the review process.

We look forward to receiving your revised manuscript.

Kind regards,

Somayeh Hessam

Academic Editor

Journal Requirements:

1. Please note that PLOS Global Public Health has specific guidelines on code sharing for submissions in which author-generated code underpins the findings in the manuscript. In these cases, all author-generated code must be made available without restrictions upon publication of the work. Please review our guidelines at https://journals.plos.org/globalpublichealth/s/materials-and-software-sharing#loc-sharing-code and ensure that your code is shared in a way that follows best practice and facilitates reproducibility and reuse.

2. In the online submission form, you indicated that "Data for the analysis is available with corresponding author upon a reasonable request". 

a. In a public repository, 

b. Within the manuscript itself, or 

c. Uploaded as supplementary information.

3. Please provide an Author Summary. This should appear in your manuscript between the Abstract (if applicable) and the Introduction, and should be 150–200 words long. The aim should be to make your findings accessible to a wide audience that includes both scientists and non-scientists. Sample summaries can be found on our website under Submission Guidelines:

https://journals.plos.org/globalpublichealth/s/submission-guidelines#loc-parts-of-a-submission.

Additional Editor Comments (if provided):

Dear authors,

Thanks for submitting your valuable manuscript to the journal. As a result of this, we would like to inform you that the review of your manuscript is finished, you need to do a "Minor Revision" on your manuscript. Even though this manuscript has enough quality to enter the review process, it needs some significant (Minor) revisions in some points of view.

Reviewers' comments:

Reviewer's Responses to Questions

**Comments to the Author**

1. Does this manuscript meet PLOS Global Public Health’s publication criteria?

Reviewer #1: Yes

Reviewer #2: Partly

Reviewer #3: Yes

2. Has the statistical analysis been performed appropriately and rigorously?

Reviewer #1: Yes

Reviewer #2: I don't know

Reviewer #3: Yes

3. Have the authors made all data underlying the findings in their manuscript fully available (please refer to the Data Availability Statement at the start of the manuscript PDF file)?

Reviewer #1: Yes

Reviewer #2: Yes

Reviewer #3: Yes

4. Is the manuscript presented in an intelligible fashion and written in standard English?

Reviewer #1: Yes

Reviewer #2: Yes

Reviewer #3: Yes

Reviewer #1: I have reviewed the original manuscript as submitted and given feedback with minor revision requests in the attached reviewed PDF copy. Overall, I find the study relevant and in compliance with the PLOS Global Public Health Journal publication standard and requirements. The statistical analysis described by the authors was conducted correctly with sound results, discussion, and conclusion sections.

Reviewer #2: The study titled “Predicting Tuberculosis (TB) with Machine Learning Approaches: An Effort to Improve Tuberculosis Diagnosis in South Africa” investigates the feasibility of using feature-based machine learning algorithms for the prediction and diagnosis of TB in South Africa. The authors compare five feature-based machine learning (ML) techniques and assess a variety of performance metrics. They also compute the most important feature predictors that contribute to each of the ML model’s predictions. The authors found that ensemble, tree-based ML methods (RFs, SGBMs) achieved the highest overall predictive performance (in terms of accuracy, sensitivity, specificity, and AUROC), though all five ML methods achieved > 70% accuracy.

Prior literature on this topic does not utilize data from low-resource settings, instead focusing on using advanced medical interventions (e.g. chest X-rays, pharmaceutical solutions) to assess the success of machine learning and deep learning in predicting TB. This work uses only non-invasively collected, self-reported personal information to predict the presence of TB.

This study’s primary contribution is its evaluation of ML for TB prediction in the South African patient population as a low-resource real-world use case. However, its applicability to other countries and regions outside of this context is limited. The high performance of the feature-based ML methods and consistency in top contributing features to prediction across these methods strengthens the study’s conclusion. The authors include a ranking of top contributing features to the prediction, which assists in understanding the model’s predictive behavior and alignment with clinical knowledge. However, this work’s conclusions is weakened by its lack of sufficient methodological details regarding rigorous evaluation, which should be clarified before considering acceptance.

Major issues

1. The authors reference other literature on using machine learning to predict TB, though prior work focuses on clinical data prevalent in high-resource medical settings (e.g. chest X-rays). The authors should comment on the availability of other feature-based machine learning studies that aim to predict/diagnose TB or, if no such work is currently available, mention this gap.

2. When referring to the hyperparameter grid search procedure (lines 240, 252, 284-285), clearly state which data subset/split was used to compute the accuracy metric used to select the best hyperparameter configuration. Standard practice computes it as an average over 5-fold cross validation folds, but please clarify. Lines 267-270 in the SGBM section defines 5-fold cross-validation. It states that the training data are split into five equal-sized subsets, but there is no mention of how these subsets are used to train/evaluate the subsequent models.

3. ANNs can have any number of hidden layers, not just three (line 245); the authors reference including the both the number of hidden layers and the number of neurons per hidden layer as tunable hyperparameters. How was the hyperparameter search conducted while varying the number of hidden layers and the number of neurons per hidden layer? Was the latter held constant while adjusting the former? Additional details on the specifics should be included.

4. Which layers of the ANN had the sigmoid activation function applied to the outputs?

5. Accuracy was reported with a 95% confidence interval. How was this interval computed? Were models re-trained on the full 70% training set and evaluated on the 30% held-out test set with bootstrapping? The confusion matrix and other metrics indicate no measure of variance, reported with a single value evaluated on the 30% held-out test set.

6. How were the top/bottom predictors for each of the ML techniques computed?

Minor issues

1. Other more robust methods to assess multicollinearity may be used, like the variance inflation factor (VIF) which calculates how much the variance of a regression coefficient is inflated due to correlations with other features. This can be more robust because it considers more than just pairwise relationships between the features.

2. Some broad terms require more grounding tomake them relevant for a technical audience (e.g. generalizability line 223, model adequacy line 289, model reliability line 289).

3. Please clarify lines 251-252: “The desired parameters for our model are for the 10-end superscript to the minus 4-end superscripts.”

4. NPV excluded from the equation definitions (lines 292-295).

5. TP and FP represent the count of positive cases, not the rate (TPR, FPR).

6. Please clarify that AUC values refer to the area under the ROC in the main text.

7. Background information on the Kappa metric, detection rate, detection prevalence, balanced accuracy would be helpful to include.

8. Accuracy > NIR was not reported for all ML techniques (SGBM, SVM excluded) in the main text.

9. DTs should not be referenced as ensemble models in lines 377-378.

10. Relative feature importance values for SVMs are not included in the main text. Please include them for completeness.

11. Table 2: the prediction and reference labels should be labeled as non-TB and TB instead of 0 and 1.

12. Table 3: the 95% CI label for DTs is excluded.

13. Table 3: ROC values are listed in the text, but should be included in Table 3 as a separate row.

14. Figure 2: axes are labeled as sensitivity and specificity but to maintain consistency with the manuscript, should be TP and FP. Keep in mind that these plots are more commonly presented with true positive rate (TPR) and false positive rate (FPR).

15. Figure 3: the relative feature importance (x-axis for RF, DT, ANN, and SGBM) ranges from 0-100 whereas it is presented on the y-axis for SVM with values < 1.0. Please make the orientation of the axes, scaling, and abbreviation methods (ANN instead of NN, SGBM instead of GBM) consistent with the manuscript.

Reviewer #3: The study which focuses on using machine learning as part of the South African National Strategic Plan for tackling HIV, STI's and TB with a particular interest in predicting TB cases in the population.

Major Comments

- In using SMOTE and ending with No TB=5,585 and TB=5,541 (approximately 1:1) where the original proportions were for No TB=35,847 and TB = 1,381 (approximately 26: 1) you may have introduced introducing biases that may have better been handled with SMOTE proportions potentially of the order of 4:1 or even matching No TB to TB before training (this may also go to the order of 4:1).

- When testing/validation was the testing/validation set separated from the original population before applying SMOTE? To avoid testing SMOTE trained data on SMOTE test data in which case the testing/validation test would not be representative of the truth in the population

- Considering the national strategic plan, in what stage of the process, would this machine learning approach fit in for South Africa? Also what part of the current strategy does this approach outperform? And what are the metrics for comparing machine learning to such a step?

Minor Comments

- Line 10-133 . Consider rewording the sentence. Its meaning is unclear. "While several strategic targets by the WHO in 2000 and 2015 have achieved their goal of halving TB (14), in the South, the confluence of TB, HIV/AIDS and increasing cardiometabolic diseases are achieving this target challenging (15)."

- The definition is reversed in 188: "High blood pressure was defined as a systolic pressure of >90mm/hg and a diastolic pressure of 140mm/hg" it should be systolic >= 140mm/Hg and diastolic >= 90mm/Hg.

- Line 199 should read "...wrongly spelled values...". Also can you explain what you mean by wrong data structures?

- From line 387 please include the name of the authors and year along with the reference "For example, studies, including the work of (5), showcase the enhanced accuracy of deep learning in classifying X-ray images of potential TB patients, facilitating early detection and subtype differentiation. Furthermore, (6) explored TB drug resistance profiles using machine learning, guiding targeted treatment strategies. Additionally, (7) employed a progressive resizing approach to train models for automatic TB diagnosis, integrating fine-tuned Normalisation-Free Networks (NFNets) and the Score-Cam algorithm to highlight specific regions in chest X-rays for detailed inference."

- Bias: How does the target variable take care of old TB cases considering the question focuses on the last 30 days?

- Is there a reason you prefer in your discussion to focus on accuracy over F1 scores considering accuracy is generally considered less reliable than F1 score?

**Do you want your identity to be public for this peer review?** For information about this choice, including consent withdrawal, please see our Privacy Policy

Reviewer #1: **Yes: ** Emmanuel Ndubuisi Egbo

Reviewer #2: No

Reviewer #3: No

---

## [Decision Letter · Decision Letter 1]

8 Apr 2025

PGPH-D-24-02823R1

Predicting and Improving Diagnosis of Tuberculosis Outcomes in South Africa Using Machine Learning Approaches

Dear Dr. Gyasi,

Thank you for submitting your manuscript to PLOS Global Public Health. After careful consideration, we feel that it has merit but does not fully meet PLOS Global Public Health’s publication criteria as it currently stands. Therefore, we invite you to submit a revised version of the manuscript that addresses the points raised during the review process.

We look forward to receiving your revised manuscript.

Kind regards,

Somayeh Hessam

Academic Editor

Journal Requirements:

1. Please note that PLOS Global Public Health has specific guidelines on code sharing for submissions in which author-generated code underpins the findings in the manuscript. In these cases, all author-generated code must be made available without restrictions upon publication of the work. Please review our guidelines at https://journals.plos.org/globalpublichealth/s/materials-and-software-sharing#loc-sharing-code and ensure that your code is shared in a way that follows best practice and facilitates reproducibility and reuse.

Additional Editor Comments (if provided):

Reviewers' comments:

Reviewer's Responses to Questions

**Comments to the Author**

Reviewer #1: All comments have been addressed

Reviewer #2: All comments have been addressed

Reviewer #4: All comments have been addressed

publication criteria?

Reviewer #1: Yes

Reviewer #2: Yes

Reviewer #4: Yes

3. Has the statistical analysis been performed appropriately and rigorously?

Reviewer #1: Yes

Reviewer #2: Yes

Reviewer #4: No

4. Have the authors made all data underlying the findings in their manuscript fully available (please refer to the Data Availability Statement at the start of the manuscript PDF file)?

Reviewer #1: Yes

Reviewer #2: Yes

Reviewer #4: Yes

5. Is the manuscript presented in an intelligible fashion and written in standard English?

Reviewer #1: Yes

Reviewer #2: Yes

Reviewer #4: Yes

Reviewer #1: Authors addressed all prior comments from reviewers. The manuscript current version as reviewed meets PLOS publication standards and is recommended for publication. Thanks

Reviewer #2: (No Response)

Reviewer #4: The paper explores an important topic in the field of public health and shows that machine learning techniques can be useful in diagnosing tuberculosis. However, to increase the validity of the results, it is necessary to provide more details about the methods and statistical analyses

1-It would be better to discuss the research background related to the use of machine learning in tuberculosis diagnosis further.

2-Details on data preprocessing, including missing data handling and normalization, are not provided.

3- The exact hyper-parameter settings for each model are not known.

4-providing confidence interval values for accuracy, sensitivity, and specificity can help to better evaluate the results.

5-Statistical comparisons between model performance have not been performed; using statistical tests could reveal significant differences.

6-Further explanation is needed about the reasons why the RF and DT models perform better than other models.

7- It would be useful to discuss limitations of the study, such as the generalization of the results to other populations.

In light of the points raised, it is suggested that the article be re-examined after making the following corrections:

- include full details about data preprocessing methods and model settings.

- Provide appropriate statistical analyses to compare model performance.

- Discuss limitations of the study and generalization of results.

**Do you want your identity to be public for this peer review?** For information about this choice, including consent withdrawal, please see our Privacy Policy

Reviewer #1: **Yes: ** Chase Emmanuel Egbo

Reviewer #2: No

Reviewer #4: No

---

## [Decision Letter · Decision Letter 2]

9 Jun 2025

PGPH-D-24-02823R2

Predicting and Improving Diagnosis of Tuberculosis Outcomes in South Africa Using Machine Learning Approaches

Dear Dr. Gyasi,

Thank you for submitting your manuscript to PLOS Global Public Health. After careful consideration, we feel that it has merit but does not fully meet PLOS Global Public Health’s publication criteria as it currently stands. Therefore, we invite you to submit a revised version of the manuscript that addresses the points raised during the review process.

We look forward to receiving your revised manuscript.

Kind regards,

Somayeh Hessam

Academic Editor

Journal Requirements:

Additional Editor Comments (if provided):

Reviewers' comments:

Reviewer's Responses to Questions

**Comments to the Author**

Reviewer #4: All comments have been addressed

publication criteria?

Reviewer #4: Yes

3. Has the statistical analysis been performed appropriately and rigorously?

Reviewer #4: Yes

4. Have the authors made all data underlying the findings in their manuscript fully available (please refer to the Data Availability Statement at the start of the manuscript PDF file)?

Reviewer #4: Yes

5. Is the manuscript presented in an intelligible fashion and written in standard English?

Reviewer #4: Yes

Reviewer #4: This manuscript presents a timely and relevant study applying multiple machine learning (ML) algorithms to improve tuberculosis (TB) diagnosis in South Africa.The study is methodologically sound in many aspects and contributes meaningfully to public health research in low-resource settings.The following are suggested for improvement:

1- Although the authors mention dataset limitations, the lack of subgroup analysis by gender or race—two key epidemiological factors in TB—is a limitation that must be better discussed.

2-The paper would benefit from further reflection on how these models could be integrated into actual clinical or public health workflows. What are the next steps for ope-rationalization?

3-Lack of baseline comparison with logistic regression:

A conventional baseline model (e.g., logistic regression) should be included for comparison to assess whether ML models truly provide superior performance.

**Do you want your identity to be public for this peer review?** For information about this choice, including consent withdrawal, please see our Privacy Policy

Reviewer #4: No

---

## [Decision Letter · Decision Letter 3]

21 Jul 2025

PGPH-D-24-02823R3

Predicting and Improving Diagnosis of Tuberculosis Outcomes in South Africa Using Machine Learning Approaches

Dear Dr. Gyasi,

Thank you for submitting your manuscript to PLOS Global Public Health. After careful consideration, we feel that it has merit but does not fully meet PLOS Global Public Health’s publication criteria as it currently stands. Therefore, we invite you to submit a revised version of the manuscript that addresses the points raised during the review process.

We look forward to receiving your revised manuscript.

Kind regards,

Somayeh Hessam

Academic Editor

Journal Requirements:

Additional Editor Comments (if provided):

Reviewers' comments:

Reviewer's Responses to Questions

**Comments to the Author**

Reviewer #4: All comments have been addressed

publication criteria?

Reviewer #4: Yes

3. Has the statistical analysis been performed appropriately and rigorously?

Reviewer #4: Yes

4. Have the authors made all data underlying the findings in their manuscript fully available (please refer to the Data Availability Statement at the start of the manuscript PDF file)?

Reviewer #4: Yes

5. Is the manuscript presented in an intelligible fashion and written in standard English?

Reviewer #4: Yes

Reviewer #4: The manuscript is scientifically sound and fits well with the scope of the journal. It would be better to pay attention to the following:

1-Extensive technical details are not necessary. Consider summarizing the model architecture and parameters in supplementary material and focus more on interpretability in the main text.It would be useful to discuss how these models are understood or used by healthcare workers in practice

2-The ethical implications of deploying ML models in healthcare, particularly with low specificity, should be further elaborated. Misclassification in TB screening may lead to over-treatment or missed diagnoses.

3-Given the role of HIV in TB comorbidity in South Africa, the absence of this variable in the dataset is a major limitation. This should be clearly emphasized and considered in the discussion

4-Interpret the results with foresight and generalizability

5-Include a figure summarizing model performance metrics side by side

**Do you want your identity to be public for this peer review?** For information about this choice, including consent withdrawal, please see our Privacy Policy

Reviewer #4: No

---

## [Decision Letter · Decision Letter 4]

9 Sep 2025

PGPH-D-24-02823R4

Predicting and Improving Diagnosis of Tuberculosis Outcomes in South Africa Using Machine Learning Approaches

Dear Dr. Gyasi,

Thank you for submitting your manuscript to PLOS Global Public Health. After careful consideration, we feel that it has merit but does not fully meet PLOS Global Public Health’s publication criteria as it currently stands. Therefore, we invite you to submit a revised version of the manuscript that addresses the points raised during the review process.

Please submit your revised manuscript by . If you will need more time than this to complete your revisions, please reply to this message or contact the journal office at globalpubhealth@plos.org. Please include the following items when submitting your revised manuscript:

We look forward to receiving your revised manuscript.

Kind regards,

Somayeh Hessam

Academic Editor

Journal Requirements:

1.  Please note that PLOS Global Public Health has specific guidelines on code sharing for submissions in which author-generated code underpins the findings in the manuscript. In these cases, all author-generated code must be made available without restrictions upon publication of the work. Please review our guidelines at https://journals.plos.org/globalpublichealth/s/materials-and-software-sharing#loc-sharing-code and ensure that your code is shared in a way that follows best practice and facilitates reproducibility and reuse.

Additional Editor Comments (if provided):

Reviewer #4:

Reviewers' comments:

Reviewer's Responses to Questions

**Comments to the Author**

Reviewer #4: All comments have been addressed

publication criteria?

Reviewer #4: Yes

3. Has the statistical analysis been performed appropriately and rigorously?

Reviewer #4: Yes

4. Have the authors made all data underlying the findings in their manuscript fully available (please refer to the Data Availability Statement at the start of the manuscript PDF file)?

Reviewer #4: Yes

5. Is the manuscript presented in an intelligible fashion and written in standard English?

Reviewer #4: Yes

Reviewer #4: The study is methodologically sound in many aspects and contributes meaningfully to public health research in low-resource settings, Fortunately, the requested improvements have been largely implementedوBut the following still need to be addressed:

1- Although the authors mention dataset limitations, the lack of subgroup analysis by gender or race—two key epidemiological factors in TB—is a limitation that must be better discussed.

2-How were the top/bottom predictors for each of the ML techniques computed?

3-Considering the national strategic plan, in what stage of the process, would this machine learning approach fit in for South Africa? Also what part of the current strategy does this approach outperform? And what are the metrics for comparing machine learning to such a step?

**Do you want your identity to be public for this peer review?** For information about this choice, including consent withdrawal, please see our Privacy Policy

Reviewer #4: No

---

## [Editor Report · Decision Letter 5]

22 Oct 2025

Predicting and Improving Diagnosis of Tuberculosis Outcomes in South Africa Using Machine Learning Approaches

PGPH-D-24-02823R5

Dear Prof Gyasi,

We are pleased to inform you that your manuscript 'Predicting and Improving Diagnosis of Tuberculosis Outcomes in South Africa Using Machine Learning Approaches' has been provisionally accepted for publication in PLOS Global Public Health.

Best regards,

Somayeh Hessam

Academic Editor